# Modulation of the Acute Inflammatory Response Induced by the *Escherichia coli* Lipopolysaccharide through the Interaction of Pentoxifylline and Florfenicol in a Rabbit Model

**DOI:** 10.3390/antibiotics12040639

**Published:** 2023-03-24

**Authors:** Victoria Cazanga, Cristina Palma, Tomás Casanova, Daniela Rojas, Karin Barrera, Cristhian Valenzuela, Aracelly Acevedo, Gabriel Ascui-Gac, Tamara Pérez-Jeldres, Rubén Pérez-Fernández

**Affiliations:** 1Pharmacology Laboratory, Department of Clinical Sciences, Faculty of Veterinary Sciences, Universidad de Concepción, Chillan 3820572, Chile; vicazanga@udec.cl (V.C.); crispalma@udec.cl (C.P.); tcasanova@udec.cl (T.C.); kbarrerac@udec.cl (K.B.); cvalenzuela2017@udec.cl (C.V.); aacevedo2017@udec.cl (A.A.); 2Veterinary Pathology Laboratory, Department of Pathology and Preventive Medicine, Faculty of Veterinary Sciences, Universidad de Concepción, Chillan 3820572, Chile; drojas@udec.cl; 3Center for Autoimmunity and Inflammation, La Jolla Institute for Immunology, California University, San Diego, CA 92182, USA; gascui@lji.org; 4Department of Gastroenterology, Faculty of Medicine, Pontificia Universidad Católica de Chile, Santiago 8331150, Chile; tamaperez@hotmail.com

**Keywords:** anti-inflammatory, cytokines, antibiotics, pentoxifylline, florfenicol, drug interactions

## Abstract

Background: Experimental reports have demonstrated that florfenicol (FFC) exerts potent anti-inflammatory effects, improving survival in a murine endotoxemia model. Considering the anti-inflammatory and immunomodulatory properties of pentoxifylline (PTX) as an adjuvant to enhance the efficacy of antibiotics, the anti-inflammatory effects of the interaction FFC/PTX over the *E. coli* Lipopolysaccharide (LPS)-induced acute inflammatory response was evaluated in rabbits. Methods: Twenty-five clinically healthy New Zealand rabbits (3.8 ± 0.2 kg body weight: bw), were distributed into five experimental groups. Group 1 (control): treated with 1 mL/4 kg bw of 0.9% saline solution (SS) intravenously (IV). Group 2 (LPS): treated with an IV dose of 5 µg/kg of LPS. Group 3 (pentoxifylline (PTX) + LPS): treated with an oral dose of 30 mg/kg PTX, followed by an IV dose of 5 µg/kg of LPS 45 min after PTX. Group 4 (Florfenicol (FFC) + LPS): treated with an IM dose of 20 mg/kg of FFC, followed by an IV dose of 5 µg/kg of LPS 45 min after FFC administration. Group 5 (PTX + FFC + LPS): treated with an oral dose of 30 mg/kg of PTX, followed by an IM dose of 20 mg/kg of FFC, and, 45 min after an IV dose of 5 µg/kg of LPS was administered. The anti-inflammatory response was evaluated through changes in plasma levels of interleukins (TNF-α, IL-1β and IL-6), C-reactive protein (CRP), and body temperature. Results: It has been shown that each drug produced a partial inhibition over the LPS-induced increase in TNF-α, IL-1β, and CRP. When both drugs were co-administered, a synergistic inhibitory effect on the IL-1β and CRP plasma concentrations was observed, associated with a synergic antipyretic effect. However, the co-administration of PTX/FFC failed to modify the LPS-induced increase in the TNF-α plasma concentrations. Conclusions: We concluded that the combination of FFC and PTX in our LPS sepsis models demonstrates immunomodulatory effects. An apparent synergistic effect was observed for the IL-1β inhibition, which peaks at three hours and then decreases. At the same time, each drug alone was superior in reducing TNF-α levels, while the combination was inferior. However, the peak of TNF-α in this sepsis model was at 12 h. Therefore, in rabbits plasma IL-1β and TNF-α could be regulated independently, thus, further research is needed to explore the effects of this combination over a more prolonged period.

## 1. Introduction

Inflammation is an essential protective mechanism against pathogen invasion and injury. During inflammation, the excessive production of pro-inflammatory molecules, such as tumor necrosis factor alpha (TNF-α) and interleukin (IL)-1β and IL-6, together with chemokines, adhesion molecules, and eicosanoids, promotes leukocyte recruitment and tissue damage. Therefore, a reduction in inflammatory mediator and cytokine production can attenuate tissue damage in infection and inflammatory diseases [1]. 

With the rate of discovery of new antibacterial drugs declining and the global increase in resistant bacteria, the ability of antibacterial drugs to modulate the immune response, either by suppression or enhancement, may be a useful therapeutic strategy to be utilized for the treatment of infectious diseases. Focus needs to be drawn to promote the use of therapies that minimize the selection of resistant bacteria and agents, which can be used alone or in combination to moderate elements of the host response to reduce the bacterial survival [2].

Florfenicol (FFC) is a fluorinated derivative of chloramphenicol, in which the hydroxyl group at C-3 has been replaced with fluorine. It is a potent inhibitor of microbial protein synthesis, which results from the inhibition of the peptidyl transferase by binding to the bacterial 50 S ribosome subunit [3,4]. It is widely used to prevent or treat bacterial infectious diseases, such as bovine respiratory disease or swine enzootic pneumonia [5]. In rabbits, FFC has been proposed as an ideal antibiotic for safe use in treating gastrointestinal and respiratory bacterial infections, such as *Escherichia coli* [6,7]. 

It has been demonstrated that some antibacterial drugs modulate the inflammatory response by affecting directly the host cell function.. Drugs that target protein synthesis or DNA replication processes in bacteria have been suspected of causing this effect [8]. These anti-inflammatory/immunomodulatory properties of antibiotics are widely used for inflammatory disorders of the respiratory tract, including asthma [9]. Experimental reports have demonstrated that FFC exerts potent anti-inflammatory effects, improving survival in a murine endotoxemia model [8,10]. Moreover, the anti-inflammatory effects of tilmicosin was demonstrated in calves infected with *Pasteurella haemolytica*, in which the antibiotic reduced the pulmonary inflammation [11]. 

Adjunctive therapy, which improves disease pathology by modulating the inflammatory response to the infection and shorten the duration of standard antibiotic treatment, is an important emerging concept in the treatment of infectious diseases. Since accumulation of cyclic adenosine 3′5′-monophosphate (c-AMP) through blocking its hydrolysis negatively regulates inflammation, several phosphodiesterase 4 inhibitors (PDE4i), has been found to be useful for treating human inflammatory diseases [12]. Pentoxifylline (PTX) is a nonspecific PDE4i that exerts an anti-inflammatory effect by increasing intracellular concentrations of c-AMP, thereby decreasing inflammatory cytokine production by mononuclear cells [13], partially through the stimulation of protein kinase A (PKA), which inhibits the production of leukotriene and TNF-α [14]. Moreover, it has been reported that prophylactic treatment with PTX effectively prevented increases in lung water, albumin lung-to-plasma ratio, markers of inflammation, and oxidative stress in models of acute lung injury [15].

In veterinary medicine, the combined use of anti-inflammatory and antimicrobial drugs represents a common clinical practice for the control of infectious processes in animals. Different anti-inflammatory/antimicrobial combinations have been tested for their pharmacokinetic and pharmacodynamic properties. In this way, the combination of antibiotics and NSAIDs has become one of the novel practical strategies in the treatment of various inflammatory processes associated with sepsis [16,17]. Some have a significant synergistic role in the control of bacterial infection and inflammatory reactions, as has been demonstrated for the combinations of ceftiofur/flunixin in pigs [16], tulathromycin/ketoprofen in cattle with BRD [17], FFC/flunixin meglumine [18], and FFC/meloxicam [19] in calves with respiratory complex diseases. However, the clinical application of NSAIDs for their anti-inflammatory effects remains controversial as they may have potential adverse effects, mainly on the gastrointestinal tract [20,21]. Gastrointestinal ulceration and bleeding have been associated with the NSAIDs-induced depression of the normal PGE_2_-induced gastric mucosal protective mechanism [21]. Thus, the co-administration of PTX and FFC could have significant effects on the inflammatory response induced by infection, suggesting that this combination has the potential to be tested as a therapeutic alternative in the treatment of bacterial infections in domestic animals. It has been hypothesized that reducing, but not fully blocking, TNF-α production by host cells would alleviate inflammatory responses and improve the outcome of antibiotic treatment during infection [12]. 

A central goal of all sepsis models is to faithfully reproduce a clinically relevant pathogenesis that is similar to the disease observed in human beings. Several clinical features are characteristic of human sepsis and are used to validate the relevance of animal models [22]; under these conditions, the LPS-induced acute inflammatory response is commonly used as an animal model of sepsis. In this way, the rabbit is the most important non-rodent laboratory animal used as an experimental model in many biological studies. It is genetically and physiologically close to humans, and, therefore, an adequate and viable model of experimental studies in the biomedical sciences [23]. Moreover, it is known that sensitivity to LPS endotoxin shows considerable differences between species, of which rodents, cats, and dogs are relatively resistant, whereas humans, rabbits, and nonhuman primates show an enhanced response [24].

The current study aims to investigate the effect of florfenicol and pentoxifylline, alone or co-administered, on changes in the secretion of inflammatory cytokines and on haematological and clinical chemistry variables induced by the intravenous administration of lipopolysaccharide of *Escherichia coli* in rabbits.

## 2. Results

### 2.1. Effects of Florfenicol and Pentoxifylline Alone or Co-Administered on Cytokine Responses Induced by LPS In Vivo

The mean plasma concentrations of TNF-α in the rabbits of the control group fluctuated between 111.1 ± 2.8 and 120.8 ± 7.7 pg·mL^−1^ from basal (T0) to 12 h after saline serum (SS) administration. The percent of change in the plasma concentrations of TNF-α from their basal values are shown in Figure 1. The intravenous administration of LPS induced a significant increase in plasma levels of TNF-α, ranging from 45.2 ± 5.0 to 119.0 ± 35% at 12 h of sampling. In the rabbits of groups 3 and 4, the administration of PTX or FFC prior to the LPS injection significantly decreased the changes in the plasma concentrations induced by the endotoxin (Figure 1). Contrary to the results observed in groups 3 and 4, the co-administration of PTX and FFC did not modify the LPS-induced increases in the plasma concentrations of TNF-α. Hence, the plasma concentrations of TNF-α, though significantly higher (*p* < 0.05) than those observed in the SS-treated group, are similar to the values observed in the LPS-treated rabbits (group 2). 

The mean plasma concentrations of IL-1β in the rabbits of the control group fluctuated between 370 ± 9 and 422 ± 19 pg·mL^−1^ from basal to 12 h after SS administration. In Figure 2, the percent of change of the IL-1β plasma concentrations from their respective basal values observed in the different experimental groups are shown. As is observed, the IV administration of LPS induced significant increases in the plasma concentrations of IL-1β, reaching a maximum percentage of change of 331.7 ± 52.3% at 3 h of sampling; these values then decreased significantly to values of 97.2 ± 15.2 and 89.0 ± 5.0% at 6 and 12 h of sampling, respectively. These values are significantly (*p* < 0.05) higher than those of the control group. The previous administration of PTX or FFC significantly decreased the rise in the plasma concentrations of IL-1β induced by LPS; however, the mean values remained elevated, reaching percentages of 114.6 ± 30.2 and 190.6 ± 46.3%, respectively. These values were significantly higher than those observed in the control rabbits. When PTX and FFC were co-administered prior to the LPS administration, the changes in the plasma concentrations of IL-1β induced by the endotoxin decreased significantly (*p* < 0.05) in comparison to the values observed in the LPS-treated group. Notably, significant differences were also observed when these values of change in the plasma concentrations of group 5 were compared with those observed in groups 3 and 4 (Figure 2).

The plasma concentrations of IL-6 in the rabbits of the control group fluctuated between 52.3 ± 0.4 pg·mL^−1^ and 50.2 ± 0.8 pg·mL^−1^. In the rabbits of group 2, significant increases in the plasma concentrations of IL-6 were observed when samples were taken at 3, 6 and 12 h after LPS administration, reaching its maximal increase of 78.6 ± 25.2% at 6 h of sampling. Similar changes were observed when the animals were previously treated with PTX (group 3) or FFC (group 4). However, although these values were lower than those observed in the LPS-treated group, the differences were not significant. Similarly, the co-administration of PTX + FFC prior to LPS injection did not modify the increase in the plasma concentrations of IL-6 induced by LPS alone (Figure 3).

The plasma concentrations of CRP are shown in Figure 4. As is observed, the basal values of CRP in the rabbits of the control group ranged from a basal value of 12.7 ± 0.5 µg·mL^−1^ to 11.9 ± 1.3 µg·mL^−1^ at 12 h after SS administration. In the rabbits of group 2, the intravenous administration of LPS significantly increased the plasma concentrations of CRP from a basal value of 10.9 ± 0.4 to 151.2 ± 88.2 µg·mL^−1^ observed at 12 h of sampling, which represents an increase of 1006 ± 357%. The administration of PTX or FFC in the rabbits of groups 3 and 4 significantly reduced (*p* < 0.05) the percent of increase in the CRP plasma concentrations induced by LPS to values of 405.7 ± 59.7 and 284.7 ± 64.4%, respectively. Similar results were observed when PTX + FFC were co-administered prior to LPS administration in the rabbits of group 5, reaching values of 125.1 ± 10.9% which are significantly (*p* < 0.05) different to those observed in the LPS-treated group (Figure 4).

Changes in body temperature are shown in Figure 5. As is observed, the IV administration of LPS significantly increased the body temperature of the rabbits (1.4 °C), an effect that was sustained for a period of 6 h, after which it decreased slowly, reaching values similar to those observed in the control group at 8 h after endotoxin administration. In the rabbits pretreated with PTX, the administration of LPS induced a significant increase in body temperature, which reached a maximum (1.3 °C) at 1 h after the injection of the endotoxin, and then declined in a biphasic mode, reaching values of 0.9 and 0.1 °C at 4 and 6 h of sampling, respectively. When FFC was administered alone or combined with PTX before LPS administration (groups 4 and 5), body temperature increased significantly, reaching its maximal values (1.2 °C: Group 4, and 1.0 °C: Group 5) at 1 h after LPS administration, and then decreased continuously, reaching values close to those of the control group (0.1 °C) at 2.5 h after endotoxin administration. The AUC_0→6h_ of temperature for control group was 232.3 ± 1.1 °C·h, while significantly higher values (*p* < 0.05) of 239.5 ± 0.6 °C·h were observed in the group treated with LPS. Values of 237.7 ± 0.6, 232.3 ± 1.9 and 230.5 ± 1.1 °C·h were obtained from the experimental groups treated with PTX + LPS, FFC + LPS, and PTX + FFC + LPS, respectively. The administration of FFC or the combination of PTX + FFC before LPS significantly decreased (*p* < 0.05) the AUC_0→6h_ values of temperature when these values were compared to those observed in the LPS-treated group.

### 2.2. Effects of Florfenicol and Pentoxifylline Alone or Co-Administered on Hematological and Clinical Biochemistry Responses Induced by LPS

In Figure 6, the changes in white blood cell count are shown. As is observed, after LPS administration in the rabbits of group 2, there were significant decreases in the total and differential white blood cell counts, an effect that was observed between 1 and 6 h after endotoxin administration, then returning to its basal values at 12 h. The administration of FFC or the combination of PTX + FFC before endotoxin administration did not modify the changes induced by LPS on the total or differential leukocyte count. However, elevated relative percentages of monocytes in the PTX + LPS group were observed at 12 h post-injection. 

In Table 1, the mean values of total proteins (TP) of the experimental groups of rabbits are shown. As can be observed, a decrease in the mean values of TP was observed at 6 after LPS administration, an effect that was significant in the groups treated with PTX + LPS, FFC + LPS, and PTX + FFC + LPS. Although the mean values of albumin and globulins showed similar tendencies to the changes observed in the values of TP, these changes were not significantly different to the basal values.

The plasma activity of hepatic enzymes ALT, AST, and GGT, and the concentrations of total bilirubin, are shown in Table 2. A significant increase in the serum activity of ALT was observed in the rabbits treated with PTX + LPS at 12 h of sampling. This result was different from the values observed in the groups treated with LPS alone, and was also different from those observed in the rabbits treated with FFC + LPS and PTX + FFC + LPS.

Significant increases in the mean values of the serum activity of the enzyme AST were observed at 6 and 12 h, in groups treated with LPS alone and in those pre-treated with PTX + LPS, FFC + LPS, or PTX + FFC + LPS prior to the intravenous administration of LPS. It is interesting to note that the highest mean AST enzyme activity was observed in the group treated with PTX, value which was significantly higher than those observed in rabbits of the control group as well as, in those treated with LPS. However, these values were not different from those observed in groups treated with FFC + LPS and PTX + FFC + LPS (Table 2).

Significant increases in the serum activity of GGT were observed at 12 h of sampling in the group of rabbits treated with LPS alone. In the group treated with FFC + LPS or PTX + LPS, the serum activity of GGT was significantly greater than that observed in the control rabbits, although this increase is lower than that of the group treated with LPS alone. No significant differences between experimental groups were observed in the mean values of the total bilirubin.

In Table 3, the mean values of the serum concentrations of creatinine and urea are shown. The statistical analysis of renal functionality indicators has shown that there were no significant changes in the values of the creatinine or urea concentrations when the data observed in the rabbits treated with LPS alone or pre-treated, either with FFC, PTX or PTX + FFC, were compared with those observed in the control.

## 3. Discussion

Systemic inflammation induced by LPS is the most well studied experimental model for bacteremia-induced fever. LPS is a component of the cell wall of Gram-negative bacteria that activates toll-like receptor 4 (TLR4). The complex responses triggered by systemic administration of LPS include the production of cytokines, chemokines, and eicosanoids by immune cells from the blood and peripheral organs [25]. Various studies have shown that the concentrations of inflammatory cytokines produced by LPS stimulation are a reliable indicator of the extent of inflammation. Therefore, inhibitors of these inflammatory molecules have been considered a potential anti-inflammatory drug [26]. Early cytokine responses after LPS challenge have been well characterized, and are known to occur within hours of LPS challenge in vivo. This first wave of cytokines has both local and systemic effects. At the site of injury, TNF-α and IL-1β activate stromal cells, including fibroblast and endothelial cells, to produce the second wave of cytokines that amplifies and propagates the inflammatory response [27]. 

TNF-α is a cytokine produced mainly by activated monocytes and macrophages, which plays a key role as a mediator of inflammation through the induction of cell death [28,29]. TNF production occurs as part of the conserved response to a wide range of cell stressors, presumably because it participates as a central mediator of the complex cytokine, hormonal and cellular signaling cascade underlying the coordinated response to invasive or threatening stimuli [30]. IL-1β shares many biological activities with TNF-α, including the induction of systemic toxicity when infused into experimental animals. Furthermore, TNF-α and IL-1β often will exhibit strong synergy with regard to cellular or organ injury [31]. 

Experimental studies have shown that interleukin 6 (IL-6) is a multifunctional cytokine with both pro-inflammatory and anti-inflammatory activity, depending on the immune response context. Macrophages are among several cells that secrete IL-6, which they express upon activation by antigens, subsequently inducing fever and production of acute-phase proteins from the liver. Moreover, IL-6 induces the final maturation of B cells into memory B cells and plasma cells, as well as an adaptive role for short-term energy allocation [32].

In the current study, we found that the inflammatory response of rabbits to LPS exposure significantly increased plasma levels of TNF-α, IL-1β and IL-6. Moreover, the pretreatment of rabbits with FFC significantly decreased the concentrations of TNF-α and IL-1β, but did not modify the increase in IL-6 levels induced by LPS. IL-6 is related to IL-1 and TNF-α in that all three of these cytokines are coordinately released from activated monocytes, and, furthermore, one can induce production of another [30]. IL-6 contributes to the inflammatory response, but also down-regulates the production of TNF-α and IL-1, thereby contributing to the limitation of the inflammatory response. IL-6 is a pleiotropic cytokine whose action depends on its local concentration, as well as the nature of the target cells. Under different circumstances, IL-6 may exert pro- or anti-inflammatory effects [26,33]. 

Non-antibiotic effects of antibiotics have been an intriguing subject in the field, but some evidence of these has recently appeared. Some antibiotics have been shown to have direct immune modulatory effects on host immune cells [34]. Interestingly, one study reported the effect of increased levels of TLR4. However, the direct reduction in the levels of IL-1β production by the effect of azithromycin has been related to the AP-1 transcription factor while NF-κB and MAPK pathways remain intact [35]. In vitro experimental data, using an LPS-stimulated RAW 264.7 macrophages model, have demonstrated that FFC reduced TNF-α and IL-6 production. Moreover, in vivo studies performed in mice challenged with LPS demonstrated that FFC significantly decreased TNF-α and IL-6 concentrations in murine serum [8]. These differences, in the inhibitory effects of FFC over the expression of IL-1β and IL-6 between a murine model [8] and this study’s lapin model of experimental LPS-induced endotoxemia, support the hypothesis that there are species differences in the expression of these cytokines. Moreover, this antipyretic effect can be associated with the results reported by [36], in an LPS-stimulated murine RAW 264.7 macrophages model, where FFC inhibits PGE_2_ production in conjunction with COX-2 expression.

As is shown in Figure 4, the IV administration of a single dose of 5 µg·kg^−1^ of LPS induced a significant increase in plasma levels of CRP, reaching a maximum of 1006% of increment from its basal values at 12 h after endotoxin injection. C-reactive protein is primarily synthetized in the liver in response to IL-6, and this synthesis is enhanced synergistically by IL-1β [37]. Evidence suggests that CRP is an important regulator of inflammatory processes, and not just a marker of inflammation or infection. Key areas of inflammation and host responses to infection mediated by CRP include the complement pathway, phagocytosis, nitric oxide (NO) release, and cytokine production [38,39]. The IM administration of a single dose of FFC significantly decreased the LPS-induced increase in plasma CRP con-centrations (Figure 4). To our best knowledge, this is the first study where the previous administration of FFC counteracted the LPS-induced increase in CRP in rabbits. The inhibitory effects of FFC on the TNF-α, IL-1β, and CRP plasma concentrations agree with the anti-inflammatory effect described in animals submitted to an LPS-induced acute inflammatory response [8,10]. 

The IV administration of LPS induced significant increases in body temperature (BT) in rabbits, an effect that was observed from 15 min after endotoxin administration, reaching a maximum increment value of 1.4 °C at 60 min after LPS injection and remaining increased (1.3 °C) for a period of 240 min, after which it decreased slowly, reaching values similar to the control group at 7 h (Figure 5). Fever is regarded as a part of the acute-phase response to infection or inflammation [40]. The fever-inducing agent that has been used most frequently in experimental studies, to characterize the physiological processes and to identify the neuronal circuit mechanisms that underline the manifestation of the febrile response, is LPS [41]. Fever is usually initiated in the periphery of the challenged host by the immediate activation of the innate immune system by LPS, specifically of the complement cascade, as this results in the immediate generation of C5a. However, LPS also activates many pathways that induce the synthesis of pro-inflammatory prostaglandins (PGs). LPS activates TLR4-inflammasome pathways, which promote the production of IL-1β, and in turn, induces the expression of cyclooxygenase (COX)-2.. This enzyme is membrane-bound and plays a key role in the synthesis of pro-inflammatory PGs. Pro-inflammatory PGs activate the NF-κB pathway in stromal cells, which in turn produce cytokines and chemokines [41] to activate and recruit neutrophils, macrophages and lymphocytes. Activated granulocytes and lymphocytes start producing TNF-α and other pro-inflammatory signals that further drive the innate immune response. 

It is generally agreed that both exogenous as well as endogenous pyrogens, when administered intravenously at relatively high doses, often induce biphasic BT rises [40,42]. The first of these two fever phases seem to be initiated prior to the appearance of a sufficient increase in pyrogenic cytokines in the blood, and anti-cytokine strategies which neutralize a LPS-induced circulating cytokine or antagonize their biological action result in attenuation of the second, but not the first, phase of the biphasic response of guinea pigs to LPS [40]. Based on studies using anti TNF-α protein and IL-1 receptor antagonist (IL-1Ra) in guinea pigs treated with LPS, it has been demonstrated that the second phase of fever can be suppressed by the administration of this antagonist, findings that suggest that circulating cytokines are not involved in the generation of the early phase of fever, but rather provide signals which contribute to the maintenance of the later phase of fever [43]. These results agree with those proposed by Blatteis [44], based on when during the course of the fever the reported data were collected. Majority of the available results seems suggest that pyrogenic cytokines in the bloodstream and in the preoptic area have roles in the later, rather than the earlier, phases of the febrile response to exogenous pyrogens [44].

As is observed in Figure 5, in the rabbits of group 4 the administration of FFC prior to LPS did not modify the first wave of increase in BT, but antagonized the sustained increase in temperature induced by LPS, as observed in the rabbits of group 2. The FFC-induced decrease in BT began at 75 min after IV administration of LPS and reached values similar to those observed in the control group at 165 min after LPS administration, and then remained under normal values for the following period of BT recording. To the best of our knowledge, there are no previous studies that report the effect of FFC on changes in the BT of animals under febrile conditions induced by LPS. These results seem to be in agreement with the partial inhibitory effects of FFC over expression of the interleukins TNF-α and IL-1β. Accordingly, based on these results, where FFC has inhibitory effects over the sustained increase in BT induced by LPS, it can be proposed that these effects are mediated through the downregulation of the cytokines TNF-α and IL-1β. 

Inflammatory reactions are multicellular events that include chemotaxis, adhesion, and controlled interactions among leukocytes, endothelial cells, and platelets on the blood vessel wall [45]. The administration of a single dose of 5 µg·kg^−1^ of LPS in rabbits in the current work induced a biphasic leukocytopenia; the first phase was extended by a period of at least 6 h, which was associated with neutropenia, lymphocytopenia, and monocytopenia. In the second phase, the leukocyte count recovered to values close to basal levels. These results are similar to those reported by van Miert et al. [46] in goats, Deldar et al. [47] in calves, and Ward et al. [48] in ponies. It has been shown that leukopenia after LPS administration is due to a faster sequestering and loss of neutrophils from the circulatory blood into the pulmonary microcirculation [48]. The decrease in WBC count in response to LPS can also be explained by mechanisms of aggregation and adherence of granulocytes and monocytes to endothelial cells, which are stimulated by the released cytokines and endothelial adhesion molecules [49]. Concerning neutropenia, margination refers to the prolonged transit of neutrophils through specific organs, which results in intravascular marginated pools; these can be found within the spleen, liver, bone marrow, and to a lesser extent in the lung [50]. Leukocyte migration through activated venular walls is a fundamental immune response that is prerequisite to the entry of effector cells, such as neutrophils, monocytes, and effector T cells, to sites of infection, injury, and stress within the interstitium. Rapid activation of endothelial cells can be induced by inflammatory stimuli, such as histamine and platelet activated factor (PAF), while slow activation can be driven by cytokines (e.g., interleukins TNF-α, and IL-1β) [51]. Considering that IL-6, as a pleiotropic cytokine, is involved in the regulation of immune responses, the acute phase response, hematopoiesis [30], and the effects observed in the leukogram are inversely related to the variations in the plasma concentration of IL-6 (Figure 3), which remains elevated for a period of 6 h and then decreases significantly at 12 h after LPS administration. 

In the current work, the previous administration of PTX, FFC or PTX + FFC did not modify the effects of LPS on the total or differential count of leukocytes; the lack of effect of these drugs, or their combination, on IL-6 expression could be related to the null effect of these drugs on the changes in the leukogram induced by the endotoxin. 

The clinical biochemistry data of the current work revealed a significant increase in the enzymatic activity of AST, observed at 6 and 12 h after LPS administration. Moreover, a slight increase in the enzymatic activity of GGT was observed at 12 h after LPS administration. Conversely, no significant changes were observed in the enzymatic activity of ALT or on total bilirubin concentrations after LPS administration. The measurement of the activity of aminotransferase enzymes ALT and AST are recommended for the assessment of hepatocellular injury, ALT being considered as a more specific and sensitive indicator of hepatocellular injury than AST [52]. 

In rabbits, it is reported that ALT is not liver-specific [53,54], however, a correlation exists between the increase in the serum activity of this enzyme and the degree of liver damage [54]. The LPS dose administered to the rabbits in the current study produces a mild acute inflammatory response (AIR). Therefore, the expected response of the liver to the dose of LPS administered to the rabbits should be a small or medium increase in hepatic enzyme activity.

In the group treated with PTX + LPS, the ALT enzymatic activity levels (U/L) were significantly higher than those observed in the control group as well as those observed in the other experimental groups (*p* < 0.05), as observed at 12 h after LPS administration. Considering that these values are significantly higher than those of the LPS-treated group, this increase in ALT enzymatic activity is attributed to the effects of PTX or its metabolites on the cellular permeability of liver cells [55,56]. 

In the rabbits treated with the combination of PTX + FFC + LPS, the AST enzymatic activity levels, although elevated over those of the control group, were lower than those of the group treated with PTX alone, and are similar to those obtained in the group treated with FFC + LPS. Hence, it is possible to conclude that the reduction of the increment in AST enzymatic activity in group 5 could be due to a partial inhibitory effect of FFC, mediated by its anti-inflammatory action [8]. Another potential explanation is that the combination of PTX + FFC + LPS diminished IL-1β levels more than the other groups. It has been reported that IL1-R1 blockage attenuates liver injury in mice models, reducing the proportion of myeloid-derived suppressor cells (MDSC) in sepsis by inhibiting their migration to the liver, as well as enhancing the bacterial clearance and promoting anti-inflammatory macrophage phenotypes [57].

It is increasingly recognized that endothelial barrier disruption in sepsis and systemic inflammation is a hallmark preceding breakdown of the microcirculation, followed by metabolic unbalance, organ failure, and death. Despite its pathophysiological relevance, a therapy to stabilize endothelial barrier disruption in septic patients is not available [58]. In addition, it is well known that the breakdown of endothelial barrier functions by bacterial LPS or proinflammatory cytokines like TNF-α, is due to a reduced c-AMP levels in endothelial cells [58]. Pentoxifylline (PTX) is a methyl xanthine derivative and a non-specific phosphodiesterase inhibitor. This drug has anti-inflammatory and antithrombotic properties, and has been used in the clinical treatment of vascular disorders to reduce blood viscosity [59,60]. It has been demonstrated that PTX increases the intracellular levels of c-AMP by inhibition of phosphodiesterase enzymes that hydrolyze c-AMP. Increased concentrations of c-AMP prevent platelet aggregation and cyclooxygenase activity, and inhibit the production of cytokines such TNF-α, IL-1β and IL-6 [61].

The oral administration of a single dose of 30 mg/kg of PTX was tolerated well by the experimental animals., and no adverse effects were observed. Contrary to this observation, Myers et al. [62] describe a transient diarrhea in pigs treated with a dose of 25 mg/kg of PTX by subcutaneous route. 

The oral dose of PTX administered to rabbits in the current study was effective at mitigating the increase in the plasma concentrations of inflammatory cytokines TNF-α and IL-1β, and on the CRP induced by LPS. Similar results were found by Sharaky et al. [63] in sheep treated with a dose of 30 mg/kg of PTX, where an inhibitory effect on the TNF increase induced by LPS was observed. However, no significant effects were observed on the increase in plasma IL-6 levels induced by LPS. These results agree with those previously reported by Zabel et al. [64] in studies performed in vivo in humans, where PTX inhibited endotoxin-induced production of TNF-α, while that of IL-6 was not affected. Similar results were reported by others, using an in vitro model of mononuclear cells stimulated with LPS [65,66]. Moreover, a meta-analysis about the effects of PTX on inflammatory markers in cardiovascular diseases showed that the anti-inflammatory effect of PTX was associated with a statistically significant reduction in the concentrations of TNF-α and CRP in plasma [67]. The mechanisms proposed to explain the inhibitory effect of PTX on the TNF plasma levels increase induced by LPS include: suppression of TNF-α gene transcription, attenuation of the response of TNF-α to LPS, and attenuation of IL-2 cytokines which stimulate TNF production [68].

In summary, the fact that the inhibitory effects of PTX on LPS-induced increases of the plasma concentrations of TNF-α, IL-1β, and CRP agree with those results described in literature demonstrates that this drug exerts significant anti-inflammatory effects, which can have therapeutic utility in the treatment of bacterial infectious diseases of animals.

During the infection, along with the process of antibiotic administration, the endotoxin, exotoxin, bacterial DNA, and cell wall components released by dead bacteria can trigger an excessive inflammatory reaction that can aggravate the condition of patients. Therefore, a viable clinical strategy would be established in order to select drugs that not only destroy bacteria but also suppress excessive inflammation [69]. 

Considering the potential anti-inflammatory and immunomodulatory properties of PTX, and its prospects as an adjuvant to enhance the efficacy of antibiotics for the recovery of the infection, the current work was designed to evaluate the anti-inflammatory and immunomodulatory effects of PTX and FFC, either administered alone or combined, on the acute inflammatory response induced by the intravenous administration of LPS. The effects were evaluated through changes in plasma levels of interleukins (TNF-α, IL-1β and IL-6), CRP, body temperature, and haematological and clinical biochemistry variables in clinically healthy adult rabbits. The results have shown that each drug can produce a partial inhibition of the LPS-induced increase in TNF-α, IL-1β and CRP. When both drugs were co-administered, a synergistic inhibitory effect on the LPS-increased plasma concentration levels of IL-1β and CRP was observed. However, the co-administration of PTX + FFC failed to modify the LPS-induced increase in the TNF-α plasma concentrations, indicating a probable antagonism between both drugs on the mechanisms that regulate the expression of TNF-α induced by LPS in rabbits. From the pharmacodynamic point of view, this antagonism could be associated with mediators that control the expression of the cytokines during the inflammatory response induced by LPS. It has been demonstrated that the transcription of TNF-α and IL-1β is mediated by the activation of the nuclear factor κ-light-chain-enhancer of activated B cells (NF-κB), which is an important mechanism for the overproduction of inflammatory mediators of macrophages in response to LPS [70]. Moreover, it has been proposed that the inhibitory activity of PDE4-inhibitors on the production of inflammatory mediators seems to be mediated via inhibition NF-κB, p38 MAP kinase, and JNK activation in macrophages [71]. Likewise, it has been demonstrated that FFC inhibits the T TNF-α expression, preventing the LPS-induced NF-κB translocation from cytoplasm into nuclear in RAW 264.7 macrophages [36]. Thus, it is probable that some competition for the activation of the NF-κB between both drugs could avoid their inhibitory effect on TNF-α expression.

Cytokines play a significant role in orchestrating immune cell activity against invading microorganisms. Indeed, some cytokines, such as TNF-α, IL-6, and IL-12, are essential for limiting bacterial growth and multiplication. However, a massive spill of these pro-inflammatory cytokines, and/or overproduction of some anti-inflammatory cytokines (e.g., IL-10), during sepsis could be detrimental. Therefore, a critical balance in cytokine concentrations is necessary to elicit the optimal response to sepsis [72]. TNF-α plays a central role in the inflammatory response that accompanies systemic inflammation. However, the complete blockade of pro-inflammatory mediators has failed to improve survival in animals and humans after septic insults; this is because the host attempts to counterbalance the exacerbated pro-inflammatory response by increasing the production of anti-inflammatory cytokines, such as IL-10. It is possible that agents that attenuate but do not completely abrogate hyper inflammation provide a clinical benefit in sepsis [73]. Pharmacological inhibition of TNF-α transcription with PTX has demonstrated a beneficial effect in the treatment of pulmonary inflammatory diseases. PTX protects rodents from acute lung injury induced by *E. coli* and staphylococcal infection, endotoxin, and TNF-α [62]. Generally, the beneficial effects of PTX therapy following TNF-α infusion or bacterial insult involve improved hemodynamic, decreased fever, and decreased serum levels of TNF-α [62]. Moreover, studies related to antibiotic uptake by human phagocytic cells have demonstrated that PTX increases the intracellular antibiotic accumulation, which could improve the bactericidal capability of phagocytic cells, thus augmenting the host response to an infectious challenge. Furthermore, phagocytes with high intracellular concentrations of an appropriate antibiotic may migrate to sites of infection and release the drug, thus serving as an antibiotic delivery system [74]. Taken together, these results seem partially validate the hypothesis proposed by Subbian et al. [12], that dampening the host inflammatory response by down-regulating TNF-α through PDE4 inhibition therapy can improve the outcome of antibiotic treatment during infection. Thus, considering that the doses used in the combination of both drugs were well tolerated by the experimental animals, and that they share common anti-inflammatory and immunomodulatory properties, their combination could represent an interesting alternative to be explored, in order to understand their antibacterial and immunomodulatory properties over a more prolonged time in an in vivo model. Another avenue to explore is the effect of this combination on antibiotic resistance and sepsis survival.

The combination of FFC and PTX in our LPS sepsis models demonstrates immunomodulatory effects. An apparent synergistic effect was observed for the IL-1β inhibition, which peaks at three hours and then decreases. At the same time, each drug alone was superior in reducing TNF-α levels, while the combination was inferior. However, the peak of TNF-α in this sepsis model was at 12 h. In future, plasma IL-1β and TNF-α could be regulated independently, and further research is needed to explore the combination’s effect over a more prolonged time. Another avenue to explore is the effect of this combination on antibiotic resistance and sepsis survival.

## 4. Materials and Methods

### 4.1. Animals

Twenty-five male and female New Zealand breed rabbits were used (6–8 months of age and 3.2–3.8 kg of body weight). Sample size was estimated according to the formula described by Eng [75]. All animals were clinically healthy and did not receive any drugs within at least one month before the beginning of the study. Throughout the study, the rabbits were housed individually at an ambient temperature of 22 ± 1 °C with a 12 h light–dark cycle, with the lights being switched on at 06:00. Experiments were conducted between 07:00 and 21:00 h. Animals were fed with pelleted feed concentrate and given free access to water and food. All procedures were performed with the authorization of the Ethical Committee for Animal Experimentation of the Faculty of Veterinary Sciences, Universidad de Concepción, Chile, Protocol CBE-12-2020.

### 4.2. Drugs and Reagents

Purified lipopolysaccharide of *Escherichia coli* O128-B12 (LPS) was purchased from Sigma-Aldrich (St. Louis, MO, USA). A stock solution of endotoxin was prepared by diluting 10 mg LPS in 10 mL of a sterile and pyrogen-free saline solution (SS). Then, one mL of the stock solution was diluted to obtain a final concentration of 20 µg/mL, necessary to give a single dose of one mL/4 kg body weight (bw). Taking as a reference the results of a previous work in rabbits treated with a low dose of 1 µg/kg [76], a medium dose of 5 µg/kg was selected according to the results obtained in a pilot experiment. A 30% injectable solution of FFC (Nuflor^®^) was purchased from Intervet Chile Ltd., under license of Schering Plough, Santé Animale Z.A., La Grindoliere—Segré). Florfenicol injectable solution was diluted in a solution that contained water, ethanol, and propylene glycol at a final concentration of 60 mg/mL to facilitate the right dose for intramuscular administration. A commercial formulation for oral administration in tablet of 400 mg/pentoxifylline was diluted in 8 mL of 5% DMSO/water to obtain a final concentration of 50 mg/mL. A topical gel formulation of 7.5% benzocaine (Doldent gel^®^, Laboratorio Maver Ltd., Santiago, Chile) for painless blood sample collection was used. 

### 4.3. Experimental Design

The rabbits were evenly distributed between the experimental groups in randomized blocks, designed considering age and weight. Blocks of animals were assigned into 5 experimental groups. 

Group 1 (Control): treated with an intravenous (IV) injection of a volume of 1 mL/4 kg bw of 0.9% saline solution (SS). 

Group 2 (LPS): treated with an IV dose of 5 µg/kg of LPS. 

Group 3 (PTX + LPS): treated with an oral dose of 30 mg/kg of PTX, followed by an IV dose of 5 µg/kg of LPS 45 min after PTX administration. 

Group 4 (FFC + LPS): treated with an intramuscular dose of 20 mg/kg of FFC, followed by an IV dose of 5 µg/kg of LPS 45 min after FFC administration. 

Group 5 (PTX + FFC + LPS): treated with an oral dose of 30 mg/kg of PTX, followed by an IM dose of 20 mg/kg of FFC and, after a period of 45 min, an IV dose of 5 µg/kg of LPS was administered.

### 4.4. Body Temperature Recording and Blood Samples Processing

Rectal temperature (RT°) was measured with a flexible thermocouple inserted in the rectum and connected to a digital thermometer (Digi Sense, Model 85-2830, Cole Palmer, Chicago, IL, USA). RT° was recorded each 15 min for a period of 5 h, then each 30 min the following time until 12 h. The area under the body temperature vs. time curve from 0 to 6 h (AUC_0→6h_) post SS or LPS administration was calculated to evaluate differences between treatments groups. Blood samples (2 mL) were collected by puncture of the auricular vein under topical anesthesia with benzocaine (Doldent gel) before (T0) and at 1, 3, 6 and 12 h after LPS or SS administration. Sterile tubes, with k3-EDTA for hematological analysis and tubes without anticoagulant for clinical biochemistry analysis, were used. The samples were centrifuged at 1200× *g* at 4 °C for 5 min to obtain plasma for determination of inflammatory markers and blood biochemistry. The concentrations of TNF-α, IL-1β, IL-6 and C-reactive protein (CRP) were determined by a rabbit-specific quantitative sandwich enzyme-linked immunosorbent assay (ELISA), using a commercial kit (TNF, IL-1β and IL-6: RayBiotech, Peachtree Corners, GA, USA) (CRP: Cloud-Clone^®^, Corp. Katy, TX, USA). Cytokins and CRP were determined according to the commercial kit instructions. The concentrations of total proteins and albumin were determined through a photometric test, according to the Biuret method for total protein, and albumin was measured by the green bromocresol method. The activity of enzymes alanine aminotransferase (ALT), aspartate aminotransferase (AST) and gamma glutamyl transferase (GGT) were determined by kinetic methods using a commercial reagent kit (Diasys Diagnostica System Gmbh, Holzheim, Germany) and measured by an autoanalyzer (Autochemistry analyzer DiRui CS-600, Changchun, Jilin, China). White blood cells and platelets counts and hemoglobin concentrations were determined by a hematological counter (Human Count 30 TS, Gmbh, Holzheim, Germany). 

### 4.5. Statistical Analysis

Results are expressed as mean ± standard error of the mean (SEM), and compared using a two-way ANOVA, associated with a Tukey–Kramer multiple comparison test, a *p*-value 0.05 was established to determine significant differences between groups. All statistical analyses were performed using Graph Pad Instat (v. 3.0; Graph Pad Software Inc., San Diego, CA, USA).

## 5. Conclusions

The combination of FFC plus PTX in our LPS sepsis models demonstrates immunomodulatory effects. An apparent synergistic effect was observed for IL-1β inhibition, which peaks at three hours and then decreases. Furthermore, each drug alone was superior in reducing TNF-α levels, while the combination was inferior. However, the peak of TNF-α in this sepsis model was at 12 h. In future, plasma IL-1β and TNF-α could be regulated independently, and further research is needed to explore the combination’s effect over a more prolonged time. Another avenue to explore is the effect of this combination on antibiotic resistance and sepsis survival.

## Figures and Tables

**Figure 1 antibiotics-12-00639-f001:**
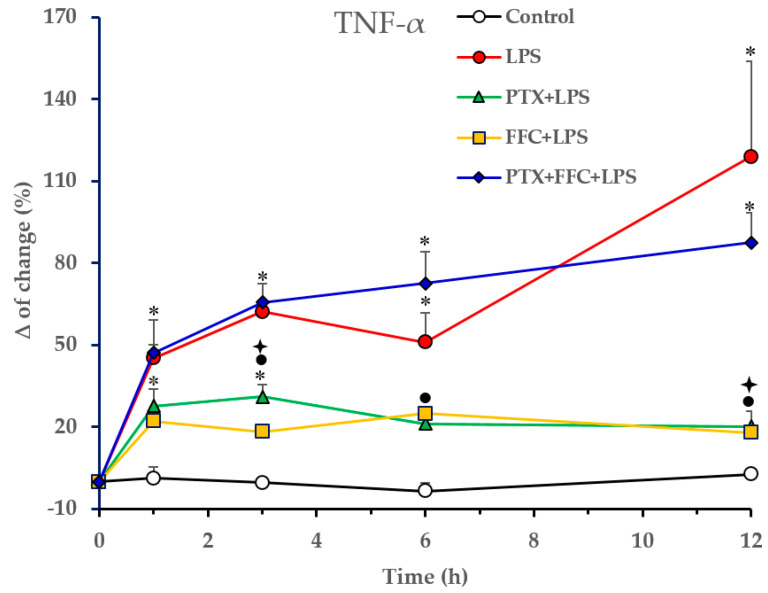
Changes in TNF-α plasma concentrations in rabbits under LPS-induced acute inflammatory response after the treatment with PTX or FFC, either alone or associated in between. Results represent the mean ± SEM of n = 5 animals per group. * *p* < 0.05 vs. control, ● *p* < 0.05 vs. LPS, 
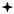

*p* < 0.05 vs. PTX/FFC/LPS.

**Figure 2 antibiotics-12-00639-f002:**
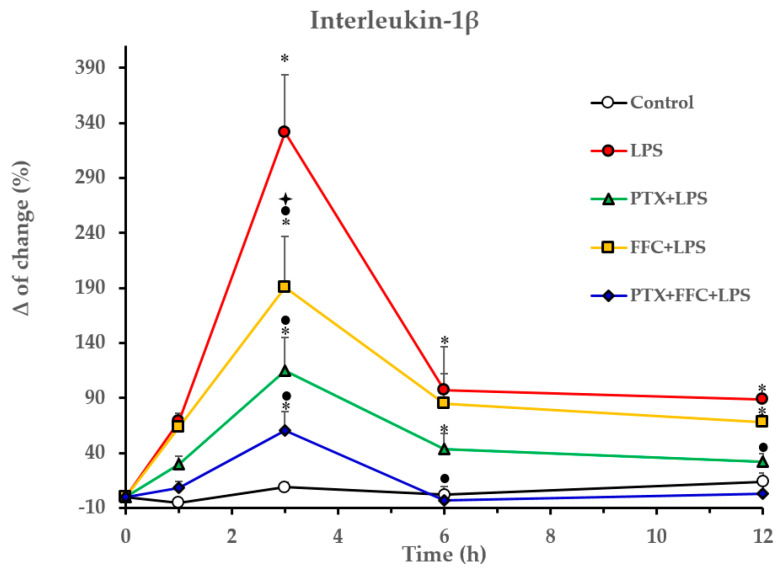
Changes in IL-1β plasma concentrations in rabbits under LPS-induced acute inflammatory response after treatment with PTX or FFC, either alone or associated in between. Results represent the mean ± SEM of n = 5 animals per group. * *p* < 0.05 vs. control, ● *p* < 0.05 vs. LPS, 
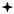

*p* < 0.05 vs. PTX/FFC/LPS.

**Figure 3 antibiotics-12-00639-f003:**
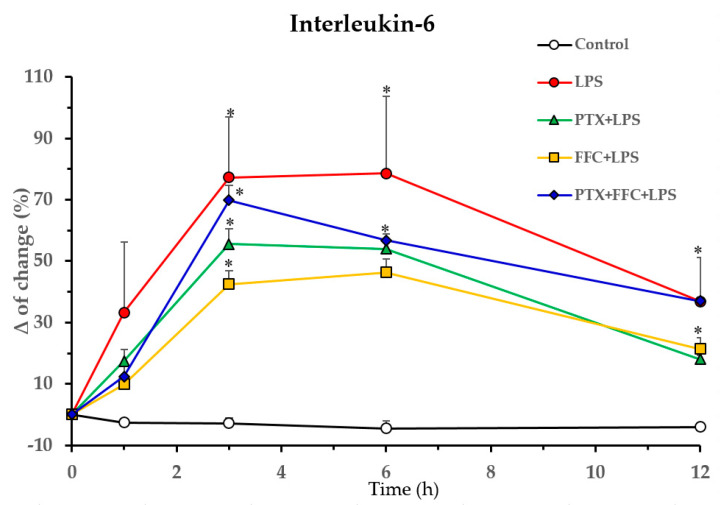
Changes in IL-6 plasma concentrations in rabbits under LPS-induced acute inflammatory response after the treatment with PTX or FFC, either alone or associated in between. Results represent the mean ± SEM of n = 5 animals per group. * *p* < 0.05 vs. control.

**Figure 4 antibiotics-12-00639-f004:**
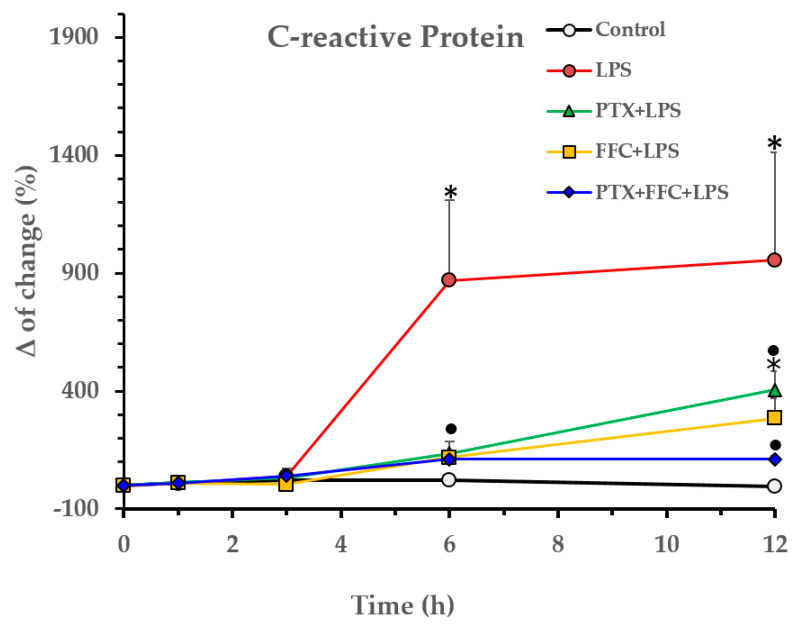
Changes in C-reactive protein plasma concentrations in rabbits under LPS-induced acute inflammatory response after the treatment with PTX or FFC, either alone or associated in between. Results represent the mean ± SEM of n = 5 animals per group. * *p* < 0.05 vs. control, ● *p* < 0.05 vs. LPS.

**Figure 5 antibiotics-12-00639-f005:**
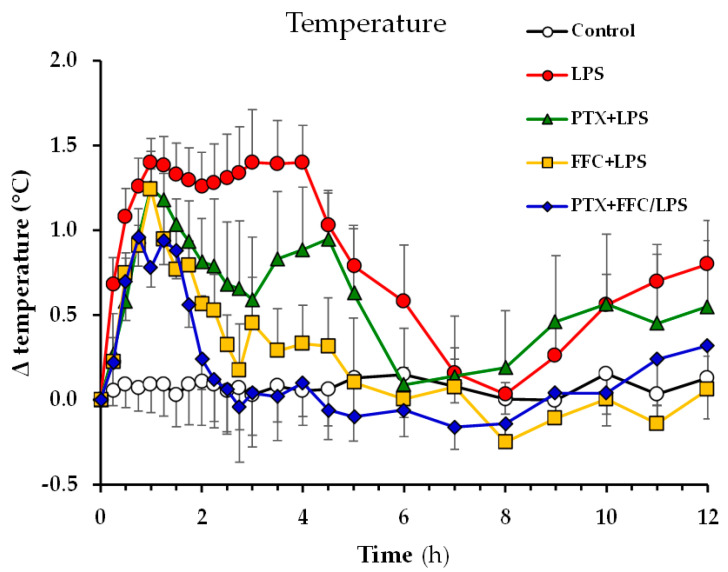
Changes in body temperature in rabbits under LPS-induced acute inflammatory response after the treatment with PTX or FFC, either alone or associated in between. Results represent the mean ± SEM of n = 5 animals per group.

**Figure 6 antibiotics-12-00639-f006:**
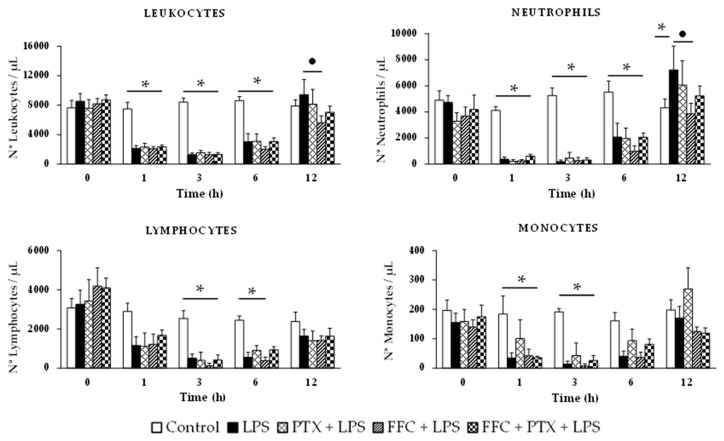
Total and differential white cells count in rabbits under LPS-induced acute inflammatory response after the treatment with PTX or FFC, either alone or associated in between. Results represent the mean ± SEM of n = 5 animals per group. * *p* < 0.05 vs. control, ● *p* < 0.05 vs. LPS.

**Table 1 antibiotics-12-00639-t001:** Mean ± SEM plasma concentrations of total proteins in rabbits under LPS-induced acute inflammatory response after the treatment with PTX or FFC, either alone or associated in between, (n = 5).

	Total Proteins (g/L)				
	0 h	1 h	3 h	6 h	12 h
Control	62.2 ± 1.4 a	58.8 ± 2.7 a	61.8 ± 2.5 a	58.6 ± 0.9 a	58.2 ± 2.2 a
LPS	61.2 ± 2.9 a	59.4 ± 3.0 a	61.2 ± 2.0 a	56.0 ± 1.7 a	56.8 ± 2.1 a
PTX + LPS	62.4 ± 3.5 a	59.2 ± 4.6 a	57.2 ± 4.4 a	53.2 ± 3.7 a *	55.8 ± 2.9 a
FFC + LPS	58.2 ± 0.5 a	55.6 ± 1.2 a	51.4 ± 1.2 b	48.4 ± 3.9 b *	50.2 ± 3.0 a *
FFC + PTX + LPS	57.0 ± 2.3 a	54.0 ± 3.5 a	50.6 ± 2.4 b	49.8 ± 2.9 b *	49.2 ± 2.2 b *

* *p* < 0.05 vs. basal value (T0); vertically different lowercase letters indicate statistically significant (*p* < 0.05) differences between groups.

**Table 2 antibiotics-12-00639-t002:** Plasma activity of hepatic enzymes aspartate alanine transferase (ALT), aspartate aminotransferase (AST), and gamma glutamyl transferase (GGT), and total bilirubin concentration in rabbits under LPS-induced acute inflammatory response after the treatment with PTX or FFC, either alone or associated in between. Results represent the mean ± SEM of n = 5 animals per group.

	ALT (U/L)				
	0 h	1 h	3 h	6 h	12 h
Control	35.2 ± 5.9 a	33.4 ± 6.4 a	38.8 ± 7.0 a	38.6 ± 5.9 a	39.2 ± 6.7 a
LPS	39.6 ± 5.9 a	37.4 ± 4.8 a	43.8 ± 3.7 a	44.6 ± 6.6 a	52.0 ± 5.2 a
PTX + LPS	46.7 ± 3.7 a	54.3 ± 7.5 a	54.0 ± 9.4 a	70.0 ± 14.0 a	148.6 ± 25.9 b *
FFC + LPS	40.6 ± 3.9 a	36.2 ± 4.1 a	36.4 ± 3.5 a	37.6 ± 7.0 a	41.0 ± 6.4 a,c
FFC + PTX + LPS	38.8 ± 4.5 a	41.8 ± 9.3 a	40.8 ± 7.9 a	56.2 ± 15.1 a	59.2 ± 17.7 a,c
	AST (U/L)				
	0 h	1 h	3 h	6 h	12 h
Control	13.8 ± 1.1 a	13.8 ± 1.5 a	19.6 ± 2.9 a	19.2 ± 2.1 a	21.8 ± 1.9 a
LPS	18.6 ± 2.9 a	19.8 ± 3.9 a	41.0 ± 6.7 a	67.6 ± 14.3 b *	77.2 ± 10.7 b *
PTX + LPS	22.0 ± 2.7 a	27.6 ± 6.0 a	48.6 ± 9.2 a	121.0 ± 27.6 b *	189.2 ± 41.6 c *
FFC + LPS	19.2 ± 3.5 a	29.2 ± 5.2 a	51.0 ± 10.5 a	69.2 ± 11.7 b *	105.0 ± 20.0 b,c *
FFC + PTX + LPS	15.4 ± 2.8 a	32.2 ± 8.1 a	49.8 ± 7.9 a	68.4 ± 10.2 b *	101.6 ± 16.3 b,c *
	GGT (U/L)				
	0 h	1 h	3 h	6 h	12 h
Control	13.4 ± 2.9 a	9.6 ± 1.0 a	10.0 ± 1.3 a	9.4 ± 1.1 a	9.6 ± 1.1 a
LPS	9.0 ± 1.5 a	9.2 ± 1.2 a	12.8 ± 1.6 a	13.4 ± 2.2 a	19.4 ± 4.0 b *
PTX + LPS	8.4 ± 1.4 a	8.8 ± 1.4 a	10.6 ± 1.4 a	11.2 ± 2.2 a	13.4 ± 2.1 b
FFC + LPS	6.8 ± 1.5 a	6.8 ± 1.9 a	9.2 ± 0.8 a	10.0 ± 0.9 a	11.4 ± 0.6 a
FFC + PTX + LPS	10.2 ± 1.3 a	10.8 ± 1.3 a	11.2 ± 1.1 a	11.6 ± 1.2 a	10.2 ± 0.8 a
	Total bilirubin (µmol/L)				
	0 h	1 h	3 h	6 h	12 h
Control	3.8 ± 0.3 a	3.8 ± 0.5 a	4.4 ± 0.6 a	4.3 ± 0.7 a	3.3 ± 0.7 a
LPS	3.7 ± 0.8 a	4.5 ± 0.8 a	4.2 ± 0.7 a	5.4 ± 1.5 a	7.1 ± 2.7 a
PTX + LPS	5.0 ± 0.7 a	4.2 ± 0.9 a	4.6 ± 0.8 a	5.2 ± 0.6 a	5.4 ± 0.8 a
FFC + LPS	3.8 ± 0.3 a	2.4 ± 0.2 a	4.0 ± 1.1 a	4.6 ± 0.7 a	3.5 ± 0.7 a
FFC + PTX + LPS	2.4 ± 0.3 a	3.3 ± 0.4 a	4.8 ± 0.4 a	7.3 ± 2.3 a	4.7 ± 0.8 a

* *p* < 0.05 vs. T0. Vertically different lowercase letters indicate statistically significant (*p* < 0.05) differences between groups.

**Table 3 antibiotics-12-00639-t003:** Mean ± SEM plasma concentrations of creatinine, urea, calcium, and phosphorus in rabbits under LPS-induced acute inflammatory response after the treatment with PTX or FFC, either alone or associated in between, (n = 5).

	Creatinine (µmol/L)				
	0 h	1 h	3 h	6 h	12 h
Control	94.4 ± 9.7 a	99.0 ± 10.3 a	97.8 ± 11.7 a	87.4 ± 10.6 a	91.2 ± 8.4 a
LPS	89.6 ± 5.6 a	90.4 ± 8.2 a	103.4 ± 6.8 a	106.2 ± 9.9 a	99.4 ± 6.5 a
PTX + LPS	81.8 ± 4.7 a	95.6 ± 5.3 a	107.4 ± 6.7 a	105.6 ± 6.1 a	99.6 ± 6.1 a
FFC + LPS	87.4 ± 5.9 a	112.0 ± 4.8 a	122.0 ± 7.9 a	130.8 ± 17.3 a	118.2 ± 11.4 a
FFC + PTX + LPS	80.4 ± 6.7 a	89.6 ± 7.0 a	102.2 ± 1.4 a	101.2 ± 5.0 a	92.6 ± 5.0 a
	Urea (mmol/L)				
	0 h	1 h	3 h	6 h	12 h
Control	6.6 ± 0.2 a	7.1 ± 0.2 a	7.5 ± 0.3 a	8.3 ± 0.4 a	7.4 ± 0.3 a
LPS	7.0 ± 0.6 a	5.7 ± 0.6 a	7.2 ± 0.8 a	8.7 ± 1.2 a	8.8 ± 1.1 a
PTX + LPS	7.7 ± 0.5 a	6.7 ± 0.6 a	7.7 ± 0.7 a	8.9 ± 1.0 a	9.5 ± 0.7 a
FFC + LPS	6.7 ± 1.1 a	5.4 ± 0.9 a	7.2 ± 0.5 a	8.8 ± 0.9 a	10.2 ± 0.8 a
FFC + PTX + LPS	6.5 ± 0.2 a	5.5 ± 0.6 a	7.1 ± 0.4 a	7.8 ± 0.1 a	8.1 ± 0.6 a

Vertically different lowercase letters indicate statistically significant (*p* < 0.05) differences between groups.

## Data Availability

The data supporting this study’s findings are available from the corresponding author upon reasonable request.

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
