# Peer review of "Modulation of the Acute Inflammatory Response Induced by the Escherichia coli Lipopolysaccharide through the Interaction of Pentoxifylline and Florfenicol in a Rabbit Model"

_antibiotics, 2023, doi:10.3390/antibiotics12040639_

Round 1

Reviewer 1 Report

The use of a multimodal approach to treating inflammatory and bacterial diseases is a challenge for humans and veterinary medicine, but a current priority due to resistant bacteria and the potential adverse effects that anti-inflammatory drugs can cause. The present article proposes a combination of pentoxifylline and florfenicol to treat a model for inflammatory reaction caused by E. coli, providing an extensive and well-structured discussion that will provide a basis for future research around both drugs. I consider that the article needs minimal changes before publishing, which I have left below. A suggestion would be to add in the introduction why did the authors decide to use a model in rabbits and not mice/rats/or other species. This is because references about sheep, humans, equines, fishes, cows, and other species were added, so it is not clear why did you use this specific animal model (e.g., perhaps, because rabbits are the principal model for inflammatory diseases, or are the only species where both drugs have been tested, etc.).   

Line 43: Write “TNF” inside the parenthesis.

Response:

Lines 64-65: I would recommend adding some examples in veterinary medicine.

Response:

Line 34: Revise the abbreviation for phosphodiesterase 4 inhibitors; it seems there is a missing “4” in “PDEi”

Response:

Line 86: Please, provide a citation where antibiotics and NSAIDs have been used together to treat sepsis.

Response:

Lines 89-91: I suggest including some examples of the gastrointestinal adverse effects that NSAIDs can cause in domestic species (e.g., Turner et al., 2006. Pharmacokinetics of meloxicam in rabbits after single and repeat oral dosing).

Response:

Figure 6: Revise the quality of the figure when uploading the final version of the article. It is blurry and the font size is too small.

Response:

Line 248 and all the abbreviations in the discussion section: Since LPS, TNF, IL, etc. have been already described in the previous sections, it is not necessary to include them here.

Response:

Lines 279-280: Please, change “but not modify the increase in IL-6…” with “but did not modify the increase in IL-6…”

Response:

Lines 502-508: It would be adequate to mention some general housing conditions of the rabbits (e.g., room temperature, relative humidity, feed, etc.).

Response:

Lines 538-539: Please, specify if the rectal temperature was assessed using a digital thermometer, including the brand, company, and country.

Response:

Author Response

Answer to reviewer 1 comments.

The authors express their appreciation to the kind collaboration given by the reviewer to the article.  All the comments and suggestions of the reviewer have been addressed and incorporated in the revised manuscript as detailed below item by item:

General Comments:

  • A new paragraph was included in the Introduction explaining why we decided to use the rabbit model. See Page 3 Lines 107-117.

Line 43. TNF was written inside the parenthesis. See line 47.

Lines 64-65: A new sentence was added giving an example of the anti-inflammatory effect of antibiotics. See lines 71-73.

Line 34 but is 37. The abbreviation for phosphodiesterase 4 inhibitors was amended as “PDE4i” See line 78.

Line 86. Two new references were included and a sentence containing examples where antibiotics and NSAIDS have been used together to treat sepsis. See line 88 and lines 94-95.

Lines 89-91. New references and a paragraph were added describing the mechanism through NSAIDs can cause gastrointestinal adverse effects in animals. See lines 98-100.

Figure 6. The Figure 6 was changed by bar graphics. See page 8.

Line 248. All the abbreviations are used through the Discussion.

Lines 279-280: The sentence was changed according to the suggested by the reviewer. See line 321.

Lines 502-508. A new para graph describing the housing conditions of rabbits was included. See lines 577-581.

Lines 538-539. A new paragraph that describes the characteristics of the thermometer used for recording rectal temperature. See lines 589-591.

Reviewer 2 Report

.* thanks for this great work but we hope to know the LPS dose depended on who (ref.)

* rewrite the Figs and tables descriptions to become clear 

* in this study you use LPS as an antigen for disease agent. but what about antibiotic resistance in natural infections? 

Author Response

The authors express their appreciation to the kind collaboration given by the reviewer to the article.  All the comments and suggestions of the reviewer have been addressed and incorporated in the revised manuscript as detailed below item by item:

  • A new paragraph was included explaining how the dose of LPS was chosen. See lines 589-591.
  • The description of Tables and figures were rewritten.
  • We understand the interest of the reviewer for the antibiotic resistance but with the LPS-induced inflammatory experimental model it is not possible to give an answer to this question.

Reviewer 3 Report

Manuscript ID: antibiotics-2269660

Type: Article

Title: Modulation of the acute inflammatory response induced by the Escherichia coli lipopolysaccharide by the interaction of pentoxifylline and florfenicol in a rabbit model.  Authors concluded that  PTX and FFC can act in opposite sense over mechanisms that regulate the expression of TNF- α induced by LPS. I have recommended Major revision.

Journal: Antibiotics 

Dear respected Editor

This work aims to explore the interactions effects of using two types of anti-inflammatory induced by LPS in rabbit’s models. This idea is innovative, with good writing

General comments

-        Please uniform of the TNF in the entire manuscript; I prefer to use TNF-α.

-        Figures should be with the format of MDPI line style (Palatino Linotype).

-        Figure 6, I invite the authors to replace another one with a good isolation. Moreover, insert the names of these figures (I mean you can write Figure 6 A-D).

-        Dissuasion needs to concise.

-         

Specific comments

Line 35-37, the conclusion should be improved. It’s not reflecting the most important results.

Line 35,  Is concluded that after??????.

Line 81-85, add citations

Line 538, please insert how to measure RT?

Line 562, when p > 0.05 and < 0.10, need to be correct.

Best regards

Author Response

Answer to reviewer 3 comments.

The authors express their appreciation to the kind collaboration given by the reviewer to the article.  All the comments and suggestions of the reviewer have been addressed and incorporated in the revised manuscript as detailed below item by item:

General Comments.

  • The abbreviation TNF-α is used throughout the entire manuscript.

-          The figures were modified according to adapt the format MDPI line style.

  •          Figure 6 was changed to a bar graphic format.
  • Discussion section  is revised

Line 35. The conclusion was modified. See lines 35-41

Lines 81-85. References were incorporated. See lines 94-95.

Line 538, please insert how to measure RT?  See line 614-616

Line 562, when p > 0.05 and < 0.10, need to be correct. See line 640

Round 2

Reviewer 3 Report

The manuscript will be of  interest for the readers of the journal and in its current status is acceptable to be published.